# Antipsychotics Affect Satellite III (1q12) Copy Number Variations in the Cultured Human Skin Fibroblasts

**DOI:** 10.3390/ijms241411283

**Published:** 2023-07-10

**Authors:** Elizaveta S. Ershova, Ekaterina A. Savinova, Larisa V. Kameneva, Lev N. Porokhovnik, Roman V. Veiko, Tatiana A. Salimova, Vera L. Izhevskaya, Sergey I. Kutsev, Natalia N. Veiko, Svetlana V. Kostyuk

**Affiliations:** Research Centre for Medical Genetics, 1 Moskvorechye St., 115522 Moscow, Russiasatelit32006@yandex.ru (N.N.V.); svet-vk@yandex.ru (S.V.K.)

**Keywords:** schizophrenia, antipsychotics, antipsychotic therapy, γH2AX, 8-oxodG, satellite III, LC3, CNV, HSF, Sat III transcription

## Abstract

The fragment of satellite III (f-SatIII) is located in pericentromeric heterochromatin of chromosome 1. Cell with an enlarged f-SatIII block does not respond to various stimuli and are highly stress-susceptible. The fraction of f-SatIII in the cells of schizophrenia patients changed during antipsychotic therapy. Therefore, antipsychotics might reduce the f-SatIII content in the cells. We studied the action of haloperidol, risperidone and olanzapine (3 h, 24 h, 96 h) on human skin fibroblast lines (*n* = 10). The f-SatIII contents in DNA were measured using nonradioactive quantitative hybridization. RNASATIII were quantified using RT-qPCR. The levels of DNA damage markers (8-oxodG, γ-H2AX) and proteins that regulate apoptosis and autophagy were determined by flow cytometry. The antipsychotics reduced the f-SatIII content in DNA and RNASATIII content in RNA from HSFs. After an exposure to the antipsychotics, the autophagy marker LC3 significantly increased, while the apoptosis markers decreased. The f-SatIII content in DNA positively correlated with RNASATIII content in RNA and with DNA oxidation marker 8-oxodG, while negatively correlated with LC3 content. The antipsychotics arrest the process of f-SatIII repeat augmentation in cultured skin fibroblasts via the transcription suppression and/or through upregulated elimination of cells with enlarged f-SatIII blocks with the help of autophagy.

## 1. Introduction

Human satellite III (1q12) (f-SatIII) is a tandem repeat located in pericentromeric heterochromatin of chromosome 1 [1,2,3,4,5], where it is collocated with satellite II sequences. Region 1q12 is characterized by high instability, chromatin rearrangements and a low level of repair of double breaks [6,7,8,9,10,11,12].

Considerable variability in f-SatIII repeat copy numbers was found earlier not only between different subjects and cultured cell lines but also between cells from the same cell population or tissue (brain). The heterogeneity of cells from the same population by f-SatIII repeat copy numbers was associated with their different stress resistance and different proliferative capacity. The lowest f-SatIII contents were found in cells derived from children’s bodies and cultured cells of early passages. Senescence and pathology are associated with an increase in repeat copy numbers [13,14,15,16].

We showed previously that cells with a high f-SatIII repeat content did not respond to the proliferative stimuli could not induce an adaptive response to damaging impacts and were highly stress-susceptible [14,17]. Thus, these cells are useless dead weight that can disrupt the functional activity of the entire cell population. This is especially true for nerve tissue cells that are involved in signal transmission along the cellular chain. The oxidative stress factors induce the death of cells with high f-SatIII repeat contents, as is the case, for example, in patients with schizophrenia [14,18]. However, using oxidative stress for the removal of non-functional cells from a tissue is fairly problematic due to the difficulty to choose the exact individual dose of exposure. As we early found, low-level stress can result in, inversely, accruing f-SatIII repeats in the cells, enriching the population with new defective cells with enlarged repeat blocks [13]. The search for compounds that block the formation and/or induce the elimination of the defective cells with a high content of f-SatIII repeats is, therefore, a challenge.

Previously, antipsychotic therapy was shown to cause changes in the repeat content in DNA isolated from blood leukocytes of schizophrenia patients. Blood cells of untreated schizophrenia patients with an initially high copy number of f-SatIII repeats showed a significant decrease in the repeat content after a course of antipsychotic therapy. At the same time, the cells of patients with initially very low f-SatIII content showed a moderate increase in the number of repeats after the therapy up to the average values [14]. Thus, antipsychotics might influence the copy number variation of f-SatIII repeats in the DNA in vivo.

This interesting ability of these compounds required a separate study in vitro. We applied a collection of 10 adult skin fibroblast (HSF) cell lines in order to explore the copy number variations of f-SatIII in the cell’s DNA under exposure to three antipsychotics widely applicable in schizophrenia therapy (haloperidol, risperidone and olanzapine).

In order to elucidate the possible mechanisms of the antipsychotics action on f-SatIII CNVs, the changes in the f-SatIII content in DNA under exposure to the antipsychotics were put together; firstly, with changes in transcription levels of these genome regions under normal conditions and under heat stress. The stress-derived transcription of the satellite repeat is assumed to cause the augmentation of the number of satellite copies. This mechanism has been studied in detail in relation to satellite II on the example of cancer cells [19]. The f-SatIII repeat is known to be transcribed in aging, cancer and irradiated cells [17,20,21]. The heat shock was previously shown to significantly stimulate the transcript formation from satellite III fragments located on chromosome 9 (9q) [22]. No similar studies for f-SatIII (1q12) fragment have been conducted ever before. It can be assumed that the CNV of satellite III is based on the same mechanism as in the case of the CNV of satellite II [19]. However, the mechanism of satellite III amplification in normal cells has not yet been studied. Secondly, the antipsychotic-induced changes in f-SatIII copy numbers in the DNA were put together with the DNA damage degree, using an oxidation marker 8-oxo-2′-deoxyguanosine (8-oxodG) and double-strand break marker histone H2AX phosphorylated on serine 139 (γH2AX). The 8-oxodG is an important biomarker of oxidative DNA damage in vivo. The 8-oxodG content reflects the oxidative stress intensity [23,24]. γH2AX is a marker of double-strand DNA breaks in human studies at the population level [25]. This protein is part of a repair complex that eliminates double-strand breaks of the DNA chain.

For the evaluation of a possible activation of genes that regulate DNA repair, apoptosis and autophagy in response to antipsychotic action, we analyzed the changes in the amounts of the corresponding proteins in the cells. The protein BRCA1 is normally expressed in the cells. It helps either repair damaged DNA or kill the cell when DNA cannot be repaired [26,27]. Proliferating Cell Nuclear Antigen (PCNA) [28] is involved in both DNA replication and DNA repair. PCNA has been shown to serve as a biomarker of DNA repair and as a processivity factor for DNA polymerase-δ that repairs the DNA gaps after the excision of damaged DNA strands [29].

B-cell lymphoma 2 (BCL2) protein regulates cell death via inhibiting apoptosis [30]. The major apoptotic activator is BCL2 Associated X (BAX) protein, which has been shown to participate actively in p53-mediated apoptosis [31]. The RNA BAX/RNA BCL2 ratio is a good apoptosis hallmark [32]. Microtubule-associated proteins 1A/1B light chain 3B (LC3) is a central protein in the autophagy pathway where it functions in autophagosome biogenesis. LC3 is the most widely used marker of autophagosomes [33].

As a result, it was shown that antipsychotics can potentially be used to reduce the number of cells with a high amount of f-SatIII repeats in the cell population. The mechanism of antipsychotic action may include an impact on the level of satellite repeat transcription and/or on the process of elimination of the defective cells by activating the autophagy process.

## 2. Results

### 2.1. Antipsychotics Reduced f-SatIII Abundance in Fibroblast DNA

HSFs of healthy controls (hc-HSFs or K1-K5) and SZ patients (sz-HSFs or SZ1-SZ5) were studied. The skin samples were obtained from SZ cases with a long-term continuous form of schizophrenia. Each patient had been administered and taking the standard antipsychotics for several years. Four of the five patients had nearest relatives with the same disease (see Section 4).The cells were exposed to drugs that are widely used for psychosis relief and stabilization of mental state in schizophrenia patients (Appendix A): Haloperidol (H), Risperidone (R), and Olanzapine (O). Using MTT [3-(4,5-dimethylthiazol-2-yl)-2,5-diphenyltetrazolium bromide] test, we chose the drug concentrations that were non-toxic for the HSFs. Appendix A shows the characteristic curve of dependence of reduced MTT form’s signal (solution absorption, λ = 570 nm) on the concentration of the drug added to the culture medium. The working (non-toxic for the cells) concentrations of drugs in the culture medium are marked with an arrow in the curve for each compound: haloperidol (1.6 µM), risperidone (7 µM) and olanzapine (5 µM). All further tests were conducted with the use of these concentrations of antipsychotic drugs, which had been added to the culture medium of subconfluent HSFs. 

The antipsychotics were added to the HSF culture medium and culture for 24 h or 96 h. Then, DNA was extracted from the cells, and the abundance of f-SatIII fragment was measured (Figure 1A(a1)). In control sz-HSFs (with no antipsychotics added), in contrast, to control hc-HSFs, the f-SatIII copy numbers increased by a factor of 1.3 to 2.3 as the duration of cultivation increased (Figure 1A(a2)). The f-SatIII repeat numbers in control sz-HSFs after 96 h of exposure exceeded the amount of f-SatIII in hc-HSFs (*p* < 0.01). Each antipsychotic drug we studied reduced the f-SatIII copy numbers in the extracted DNA after both 24 h and 96 h of exposure (Figure 1A(a3),C).

In order to analyze the effect of antipsychotics on the amount of f-SatIII in HSFs under heat shock, the subconfluent cells were heated at 42 °C for one hour. The antipsychotic was added to the culture medium 30 min before cell heating started. After the heat shock impact had been relieved, the cells were cultured for three more hours, then the repeat copy numbers were measured in the extracted DNA (Figure 1B(b1)). The heat shock induced a decrease of the f-SatIII content in DNA extracted from the control cells (Figure 1B(b2)) compared to the cells, which had been cultured without a heat shock. The antipsychotics induced an additional reduction of the repeat count in the DNA under conditions of heat shock (Figure 1B(b2,b3),C) compared to normal cultivation.

Ribosomal repeat (rDNA) was applied as a control repeat. It was shown earlier that the content of this repeat in the cells did not depend on the culture conditions [13,15,16,17]. Cell cultivation under normal conditions and heat stress had practically no effect on the rDNA abundance in the extracted DNA. Antipsychotics had no effect on rDNA fraction size in the cell’s DNA as well (Figure 1D).

Thus, we showed that exposure to antipsychotics drugs haloperidol, risperidone and olanzapine, which had been added to the HSF culture medium at a non-toxic dose, resulted in the reduction of the content of f-SatIII satellite repeats in the cell populations under the conditions of both normal cultivation and heat shock.

### 2.2. Antipsychotics Reduced the Contents of 8-oxodG in Fibroblast DNA

Earlier, it was established that f-SatIII CNVs in primary and cultured cells were associated with changes in the level of oxidative stress in the cell population [13]. Figure 2 presents the data relevant to the contents of DNA oxidation marker 8-oxodG in extracted DNAs, for which f-SatIII contents were measured as well. An enzyme immunoassay with the use of antibodies to 8-oxodG conjugated with alkaline phosphatase (Appendix A) was applied for the quantification of the oxidation marker. The data for all HSFs are shown in Figure 2A. HSFs cultivation without an exposure to an antipsychotic drug was followed by an increase in the DNA oxidation level (Figure 2A(a2)). The antipsychotics induced a decrease in the oxidation level of DNA isolated from the HSFs, which had been cultivated for 24 h or 96 h.

We analyzed a link between 8-oxodG level and f-SatIII repeat content in the DNAs isolated from HSFs (Appendix A) for all the data (24 and 96 cultivation hours in the presence and in the absence of antipsychotics, *n* = 80). A positive linear correlation between the parameters was observed. The more oxidation marker 8-oxodG found in cellular DNA, the higher the f-SatIII repeat count. The curve slope was different for different sz-HSFs and hc-HSFs. A subgroup that included SZ2 (4d), SZ4 (4d), K3 (4d) and K5 (4d) differed from a subgroup that included all the other HSFs by a higher curve slope.

Two variants of explanation of CNVs of f-SatIII and 8-oxodG in DNA extracted from cells exposed to antipsychotics can be suggested. Either the marker content can change in the same way in all the cells of the population, or the marker content can substantially vary in a small fraction of cells, which is considerably different from the other cells by the marker content. We, therefore, estimated the variability of the HSFs populations by the content of 8-oxodG using the FCA technique (Figure 2B).

The content of oxidation marker 8-oxodG in HSFs was assayed after 24 h of exposure to the drugs. Each HSF population contained two subpopulations of the cells (fractions R1 and R2), which differed by the 8-oxodG content (Figure 2B(b1)). Fraction R1 (8-oxodG+) encompassed 4% to 8% of the total cells of the HSF population. The size of these fractions became considerably reduced after exposure to antipsychotics (Figure 2B(b2,b3)). The content of 8-oxodG in the cells of major fraction R2 did not significantly change in most HSF populations (Figure 2B(b4)).

Thus, the decrease in the content of oxidation marker 8-oxodG in DNA extracted from the cells exposed to antipsychotics appeared to be associated with a reduction of the size of a small fraction of cells with a high amount of this marker.

The reduction of the number of cells with high DNA oxidation degree upon exposure to the antipsychotics can be caused by two factors. First, the antipsychotics might activate the DNA repair processes resulting in a decrease in the DNA oxidation degree. Second, the drugs might induce elimination from the population (death) of the cells with damaged DNA also resulting in a decrease in the 8-oxodG content in the total isolated DNA. In order to understand whether the process is really taking place, we have analyzed the expression of some genes responsible for the repair and elimination of damaged cells from the HSF population.

### 2.3. The Effect of Antipsychotics on the Levels of Proteins Involved in DNA Damage Repair

We analyzed the effect of antipsychotics on the level and distribution of protein γH2AX in the cell populations using FCA (Figure 3A). The control HSF populations contained low amounts of cells with high γH2AX expression. These cells form fraction R1 (Figure 3A(a1)). Under exposure to antipsychotics (24 h), the size of fraction R1 decreased (Figure 3A(a2,a3)). Meanwhile, the amount of protein γH2AX in fraction R2 practically did not change (Figure 3A(a4)).

Protein BRCA1 is part of the DNA repair complex [26,27]. Exposure to antipsychotics (24 h) practically did not change the level of this protein in the cells (Figure 3B). Protein PCNA participates in the process of excision repair of single-strand DNA breaks [29]. Upon exposure to antipsychotics, the amount of this protein (Figure 3C) either held constant or increased (SZ1, SZ5, K2). 

Thus, studying the influence of antipsychotics on subconfluent cells, we have not found any pronounced signs of changes in the levels of proteins involved in DNA repair. At the same time, we have observed a decrease in the fraction of cells with a high content of the marker of double-strand breaks γH2AX.

### 2.4. The Effect of Antipsychotics on the Levels of Proteins Involved in the Process of Elimination of Damaged Cells from the Population

Figure 4 shows data on the effect of antipsychotics on the expression of two genes that regulate apoptosis. After adding the drugs to the culture medium (in 3 h and 24 h), we observed a decrease in the ratio between the amounts of RNAs of proapoptotic gene BAX and antiapoptotic gene BCL2 (Figure 4A). The RNA *BAX* to RNA *BCL2* ratio is one of the apoptosis markers [32]. 

After an exposure to antipsychotics, a decrease of BAX protein level was also observed (Figure 4B); moreover, first of all, the number of cells with a high level of this protein expression decreased.

After an exposure to antipsychotics for 24 h, the level of autophagy marker protein LC3 increased significantly (Figure 5A). The high LC3 level was registered in most cells of the population, regardless of the amount of the DNA oxidation marker in these cells (Figure 5A(a2)). We found a negative correlation (Rs = −0.57 *p* < 0.001) between the level of protein LC3 in the cells after 24 h of antipsychotic exposure and the count of f-SatIII repeats in the DNA isolated from the cells, which were cultivated for 96 h (Figure 5B) in the presence of the drugs. The more protein LC3 was contained at the beginning of cultivation, the less copies of f-SatIII repeats accrued in the isolated DNA. Populations with a low level of protein LC3 carried the minimum amounts of the repeat in the extracted DNA.

Thus, the antipsychotics lowered the apoptosis intensity in the cell population, with inducing at the same time the process of autophagy.

### 2.5. The Influence of the Antipsychotics upon the Level of RNASATIII in the HSFs

It is assumed that one of the components of the mechanism that causes CNVs of the repeats of the centromeric satellite is transcription of these DNA regions resulting in the biosynthesis of lncRNA (long non-coding RNA) [19]. We studied the changes in RNA*SATIII* amounts in HSFs, which had been exposed to the antipsychotics for 3 and 24 h (Figure 6A). During the cultivation of control cells, we observed changes in the RNA*SATIII* contents in RNA samples obtained from HSFs (Figure 6A(a2)). In 8 HSFs, the amounts of RNA*SATIII* increased during cultivation by a factor of 1.3 to 138. In SZ1 and K2, the RNA*SATIII* contents decreased in RNA samples. 

Each antipsychotic induced a decrease of RNA *SATIII* amount in 3 h after it had been added to the culture medium. In 24 h, the RNA amount was also decreased in all HSFs, except for SZ1 and K2. In those HSFs, the antipsychotics induced an increase of the satellite transcript amount after 24 h of exposure.

The heat shock induced a decrease in RNA*SATIII* level in the control cells (Figure 6B(b1,b2)). Under the conditions of heat shock, the antipsychotics decreased the amount of RNA*SATIII* by a factor of 1.5 to 10 (Figure 6B(b3)).

It was shown a positive correlation between the f-SatIII abundance in the cells, which had been cultured for 96 h (in both the presence and absence of the antipsychotics), and the levels of RNA*SATIII* in RNA isolated from cells, which had been cultured for 3 h after adding the antipsychotic drug (Appendix A). Another positive correlation was found between the amounts of RNA and f-SatIII for the cells, which had undergone heat shock in the presence and in the absence of the drugs (3 h long cultivation after heating). 

Thus, the antipsychotics reduced the level of RNA*SATIII* in the cell’s RNA during cultivation in both normal conditions and the conditions of heat shock.

## 3. Discussion

We studied the influence of three antipsychotics on the CNVs of f-SatIII, which is part of chromosome 1 (region 1q12) pericentromeric heterochromatin in cultured skin fibroblasts derived from adult patients with schizophrenia and healthy controls.

The cultured fibroblasts are heterogeneous in terms of f-SatIII repeat content in the cells that belong to the same population. Previously, it was shown that the cells with a small number of f-SatIII copies demonstrated the maximum proliferative potential [14]. Cells with a high repeat content had a low proliferative potential and accrued in the HSF population during replicative aging. A high content of f-SatIII in the nuclei has also been shown to associate with impaired cell response to proliferative and other stimuli. Such cells are not able to develop an adaptive response, for example, under the action of low doses of radiation or other factors inducing oxidative stress. The inability to respond to the stimuli is associated with the lack of ability to remodel the architecture of chromatin, which is required to change the genome expression profile in response to stress [17]. The cells with an enlarged block of pericentromeric heterochromatin, which harbors the f-SatIII arrays, are sensitive to the action of ROS and die at a relatively low level of oxidative stress, while cells with a low count of the repeats develop an adaptive response under the same conditions [14,15,17].

Previously we proposed a “pendulum” model, suggesting a hypothetical mechanism, which provides for SatIII copy gain during the stress response, alongside the other, reverse mechanism that might reduce the mean SatIII copy number, likely via the selection of cells with excessively large 1q12 loci. Both mechanisms, working alternatively like swings of the pendulum may ensure the balance of SatIII copy numbers and optimum stress resistance [13]. In this study, we discovered some facts that corroborate and detail the model proposed earlier, and we supplemented our knowledge with new facts about chemical compounds (drugs) that aggravate the reverse mechanism to eliminate the cells with very large 1q12 loci and thus reduce the SatIII repeat copy number after a period of copy gain, leading to the copy number normalization. In particular, we have learned that the antipsychotics lower the apoptosis intensity in the cell population while inducing at the same time the process of autophagy and interrupting the process of copy gain by blocking satellite repeat transcription.

Therefore, the principal findings of this study are the following: exposure to antipsychotics results in a reduction of f-SatIII abundance in HSFs (Figure 1). The antipsychotics may be considered as agents that accelerate the elimination of “bad” cells (cells with damaged DNA, Figure 2 and Figure 3) from the population and/or allow preventing the formation of such cells. 

The mechanism of antipsychotic action on the f-SatIII content remains unknown. One can only make some speculative assumptions, which are illustrated by a scheme presented in Figure 7. An increase in the f-SatIII content in a fraction of the cells of the population is associated with the transcription from satellite repeats, which is launched when the heterochromatin–euchromatin balance is disturbed by shifting towards euchromatin [19]. The cells that produce RNA*SATIII* seem to appear due to a local disturbance of cellular homeostasis under the oxidative stress-inducing impacts. Previously, we found that an increase in the f-SatIII content in the cellular DNA negatively correlated with a decrease in the telomere repeat count [14,15,16]. The telomere shortening is associated with aging and/or oxidative stress and is accompanied by decondensation of heterochromatin sites [34,35]. RNA *SATIII* is a matrix for DNA synthesis conducted by reverse transcriptase and resulting in the formation of RNA-DNA hybrids, which insert in the chromatin site carrying the satellite repeats [19].

Antipsychotics block the synthesis of RNA*SATIII* in the exposed cells within the first hours after being added to the culture medium (Figure 7). A reduction of RNA*SATIII* amounts was also observed in drug-exposed cells after heat stress. The causes of blocking the synthesis of RNA*SATIII* by antipsychotics are still unclear. Perhaps, these compounds interact with DNA and stabilize heterochromatin or affect the process of satellite DNA transcription itself.

Another possible alternative or parallel mechanism leading to a decrease in the fraction of cells with a high content of f-SatIII upon an exposure to antipsychotics is their elimination accelerating. The cells with large f-SatIII blocks accrue DNA lesions due to the lack of an adaptive response to various DNA-damaging stimuli [17]. We have found that antipsychotics upregulate the expression of LC3 protein, a key participant in the autophagy process. An induction of the autophagy can result in the elimination of the cells with a large number of f-SatIII (and DNA lesions) from the population. Antipsychotics were recently shown to induce the processes of autophagy in the cells. In schizophrenia, autophagy processes are disrupted, and the authors believe that the induction of autophagy by some antipsychotics makes it possible to eliminate damaged brain cells from the areas associated with impaired thinking [36,37,38,39,40,41]. One can assume that such damaged brain cells belong to the fraction of cells with a high content of f-SatIII repeats, which cannot function normally and are eliminated. Antipsychotics inducing autophagy have been even suggested to be used as additional drugs in cancer treatment [42,43]. To summarize, we could anticipate a wider scope of application for antipsychotic drugs and similar chemicals in the future, not restricted by the field of psychiatry.

## 4. Materials and Methods

### 4.1. Experiment Design

HSFs of healthy controls (hc-HSFs, K1-K5) and SZ patients (sz-HSFs, SZ1-SZ5) were studied. The cells were exposed to drugs that are widely used for psychosis relief and stabilization of mental state in schizophrenia patients (Appendix A): Haloperidol (H), Risperidone (R) and Olanzapine (O). Using MTT test, we chose the drug concentrations that were non-toxic for the HSFs. Figure 1 shows the characteristic curve of dependence of reduced MTT form’s signal (solution absorption, λ = 570 nm) on the concentration of the drug added to the culture medium. The working (non-toxic for the cells) concentrations of drugs in the culture medium are marked with an arrow in the curve for each compound: haloperidol (1.6 µM), risperidone (7 µM) and olanzapine (5 µM). All further tests were conducted with the use of these concentrations of antipsychotic drugs, which had been added to the culture medium of subconfluent HSFs. The time periods of cell exposure to antipsychotic drugs were 3 h, 24 h (1 d) and 96 h (4 d) in different tests.

### 4.2. Participants

Primary adult HSFs of healthy controls (*n* = 5) and schizophrenic (SZ) patients (*n* = 5) were obtained from the collection of the Research Centre for Medical Genetics. A description of the five healthy subjects and the five schizophrenic patients, who were the primary donors of skin cells, is presented in Table 1. The examination was conducted in the conditions of round-the-clock inpatient units of the Clinical Psychiatric Hospital #14 (N.A.Alekseev Clinical Psychiatric Hospital branch), where the patients received standard psychiatric care. The skin samples were obtained from SZ cases with a long-term continuous form of schizophrenia. Each patient had been administered and taking the standard antipsychotics for several years (haloperidol, aminazin, risperidone, seroquel and olanzapine). Four of the five patients had nearest relatives with the same disease.

The control skin sample donors were staff members of the Research Centre for Medical Genetics. They were apparently healthy, took no pharmaceuticals and had no mental problems and relatives with mental disorders.

### 4.3. Ethical Approval for the Use of Cultured Human Cells

The study was carried out in accordance with the latest version of the Declaration of Helsinki and was approved by the Independent Interdisciplinary Ethics Committee on Ethical Review for Clinical Studies (Protocol #4 dated 15 March 2019 for the scientific minimally interventional study “Molecular and neurophysiologic markers of endogenous human psychoses”). All participants signed an informed written consent to participate in the study after the procedures had been completely explained.

### 4.4. Cell Cultures and Drugs

The cells were maintained in Dulbecco’s Modified Eagle Medium supplemented with 10% fetal bovine serum (HyClone Laboratories, Logan, UT, USA), 2 mM glutamine, 50 U/mL penicillin and 50 μg/mL streptomycin at 37 °C in a 5% CO_2_ incubator. All the experiments were performed using cells from passage 4. Cells with less than 10% S-phase were considered a subconfluent culture. 

The drugs that are widely used to stabilize the mental state in patients with schizophrenia were investigated (Figure 1): Haloperidol (H), 4-[4-(4-chlorophenyl)-4-hydroxypiperidin-1-yl]-1-(4-fluorophenyl) butan-1-one; Risperidone (R), 3-[2-[4-(6-fluoro-1,2-benzoxazol-3-yl)piperidin-1-yl]ethyl]-2-methyl-6,7,8,9-tetrahydropyrido [1,2-a]pyrimidin-4-one; Olanzapine (O), 2-methyl-4-(4-methylpiperazin-1-yl)-10H-thieno [2,3-b][1,5]benzodiazepine.

### 4.5. DNA Isolation from HSFs

A total of 5 × 10^5^ cells were placed in a well of a six-well plate, cultured for 24 h and drugs were added. The cells were then cultured 3–4 h, 24 h (1 day) or 96 h (4 day). In order to isolate DNA, we used the standard method described in detail previously [34]. Briefly, 2 mL of the solution (2% sodium lauryl sarcosylate, 0.04 M EDTA and 150 μg/mL RNAse A (Sigma, St. Louis, MO, USA)) was added to the cell mass for 45 min (37 °C) and then treated with proteinase K (200 μg/mL, Promega, Madison, USA) for 24 h at 37 °C. The lysates were extracted with an equal volume of phenol, phenol/chloroform/isoamyl alcohol (25:24:1) and chloroform/isoamyl alcohol (24:1), respectively. DNA was precipitated by adding 1/10 of a volume of 3 M sodium acetate (pH 5.2) and a 2.5 volume of ice-cold ethanol. DNA was collected by centrifugation (10,000× *g* for 15 min at 4 °C), washed with 70% ethanol (*v*/*v*) and dissolved in water. DNA concentration was measured in two steps, as described earlier [34]. The final DNA quantification was performed fluorimetrically using PicoGreen dsDNA quantification reagent (Invitrogen, Carlsbad, CA, USA). DNA concentration in a sample was calculated according to a DNA standard curve. We used EnSpire equipment (PerkinElmer, Inc. Waltham, MA USA) at λex = 488 nm and λem = 528 nm.

### 4.6. Nonradioactive Quantitative Hybridization (NQH)

NQH method is based on the complementary hybridization of DNA immobilized on a filter with a biotin-labeled DNA probe. The NQH was used for the quantification of f-SatIII, TR and rDNA repeats, as specified in detail previously [44]. We made no modifications to the technique described. Five reference DNA probes with a known repeat content were applied on the same filter. After hybridization and biotin detection using the streptavidin-alkaline phosphatase conjugate (BCIP and NBT substrates), the filter was scanned and the spot signal intensity was measured using a customized software pack. A calibrate curve was built, reflecting the dependence of signal intensity on the genome’s repeat copy number. The relative standard error for NQH was merely 5 ± 2%. The major overall error of the experiment was contributed by the step of isolating DNA from the cells. The total standard error was 11 ± 7%.

DNA Probe for SatIII (1) quantification was a 1.77-kb cloned EcoRI fragment of human satellite DNA [2] labeled with biotin-11-dUTP using nick translation. Dr. H.Cook (MRC, Edinburgh, UK) kindly supplied the human chromosome lql2-specific repetitive satellite DNA probe pUC1.77.

DNA Probe for rDNA quantification contains rDNA sequences (5836 bp) cloned into an EcoRI site of the pBR322 vector. The rDNA fragment covered positions from −515 to 5321 of the human rDNA (GenBank accession No. U13369).

### 4.7. Flow Cytometry Analysis (FCA)

Cells were analyzed at CytoFLEX S (Beckman Coulter, Brea, CA, USA). HSFs were grown in 60 mm dishes. Prior to FCA, cells were washed with Versene solution and treated with 0.25% trypsin under light microscope observation. Then the cells were transferred to Eppendorf tubes, washed with culture medium, centrifuged and resuspended in PBS.

Staining the cells with antibodies was performed as described previously [45]. Briefly, to fix the cells, PFA (Sigma, Kawasaki, Japan) was added (3% at 37 °C for 10 min). Then the cells were washed 3 times with 0.5% BSA-PBS and permeabilized with 0.1% Triton X-100 (Sigma) in PBS. Cells (~50 × 10^3^) were stained with the following antibodies: 8-oxodG-PE (sc-393871 PE, ‘Santa Cruz Biotechnology, Inc.’, Santa Cruz, CA, USA), γH2AX-pb450 (nb100-384AF405, ‘Novus Biologicals, LLC’, Centennial, CO, USA), BAX-PE (rabbit Nb120-7977, NovusBio, Centennial, CO, USA), secondary mouse anti-rabbit IgG-PE (sc-3753 Santa Cruz Biotechnology, USA), A350-BCL2 (bs-15533r-a350, Bioss Antibodies Inc. Woburn, MA, USA) and LC3DyLight 488 (NB100-2220G, NovusBio, USA). To quantify the background fluorescence, we stained a portion of the cells with secondary FITC (PE, pb450, DAPI)-conjugated antibodies only. Primary data are presented as median signal intensities minus signal background values. For a more convenient presentation of the data, the values of the medians of FL-signal were normalized to the maximum FL-signal value in the sample (*n* = 10). The relative standard error of the FCA was 4 ± 2%.

### 4.8. Real-Time PCR

Isolation of the RNA was performed using a RNeasy Mini kit (‘Qiagen’, Hilden, Germany). RNA was quantified using a Quant-iT RiboGreen RNA reagent dye (‘MoBiTec’, Göttingen, Germany) on a reader (EnSpire equipment PerkinElmer, Inc. Waltham, MA USA) at λem = 487 nm, λfl = 524 nm. After DNAse I treatment, RNA samples were reverse transcribed using a Reverse Transcriptase kit (‘Sileks’, Moscow, Russia). PCR was conducted with the specific primers and Sybr-Green intercalating dye on a StepOnePlus device (‘Applied Biosystems’, Foster City, CA, USA). The primers were selected and synthesized by ‘Evrogen’ (Moscow, Russia). The internal standard was TBP gene. For the RNASATIII assay, primers described in [21] were applied.

The PCR reaction mixture in a volume of 25 µL consisted of 2.5 µL PCR buffer (700 mMol/L Tris-HCl, pH 8.6); 166 mMol/L ammonium sulfate, 35 mMol/L MgCl2, 2 µL 1.5 mMol/L dNTP solution; 1 µL 30 pMol/L primer solution, cDNA. PCR conditions were chosen individually for each primer pair. After denaturation for 4 min at 95 °C, 40 amplification cycles were performed in the following order: 94 °C for 20 s, 56–62 °C for 30 s, 72 °C for 30 s, and 72 °C for 5 min. The data were processed using a calibration plot with a resultant error of 2%.

### 4.9. MTT Test

Cells were grown in 96-well plate for 48 h. The cells were then incubated for another 72 h in the presence of various amounts of antipsychotics. Eight parallel wells of the plate were used for each antipsychotic concentration (SD is given). The survival analysis was conducted using 3-(4,5-dimethylthiazol-2-yl)-2,5-diphenyltetrazolium bromide (MTT) as described earlier [46]. The plates were read at 570 nm using EnSpire reader.

### 4.10. DNA Oxidation Analysis in HSFs

The method for 8-oxodG quantification was specified in detail previously [47]. Briefly, the DNA samples were applied to a prepared filter (Optitran BA-S85, GE healthcare, Chicago, IL, USA). Three dots (10 ng/dot) were applied per sample. Five standard samples of the oxidized genomic DNA (10 ng/dot) with a known content of 8-oxodG (was determined by ESI-MS/MS using AB SCIEX 3200 Qtrap machine) were applied to the same filter, in order to plot a calibration curve for the dependence of the signal intensity on the number of 8-oxodG in a particular DNA sample. The filter was heated at 80 °C in vacuum for 1.5 h. 8-oxodG antibody conjugated with alkaline phosphatase was used. Then the filter was placed into a solution of substrates for alkaline phosphatase NBT and BCIP. Upon the completion of reaction, the filter was washed with water and dried in the darkness. The dried filter was scanned. For the quantitative analysis of the dots, special software was used (Images, version 6, RCMG, Moscow, Russia). Signals from several dots for the same sample are averaged. The 8-oxodG content in a studied sample is calculated using the calibration curve equation. Relative standard error was 15 ± 5%.

### 4.11. Statistical Analysis

Each test was repeated in triplicate. In FCA, the medians of the signal intensities were analyzed. The significance of the observed differences was analyzed with the nonparametric Mann–Whitney U test. The *p*-values < 0.01 were considered statistically significant. The data were analyzed with Microsoft Excel and Office 2013 (Microsoft, Redmond, WA, USA), StatPlus2007 Professional software (http://www.analystsoft.com accessed on 19 March 2022) and STATGRAPHICS® Centurion XV (Statpoint Technologies, INC., Plains, VA, USA).

## Figures and Tables

**Figure 1 ijms-24-11283-f001:**
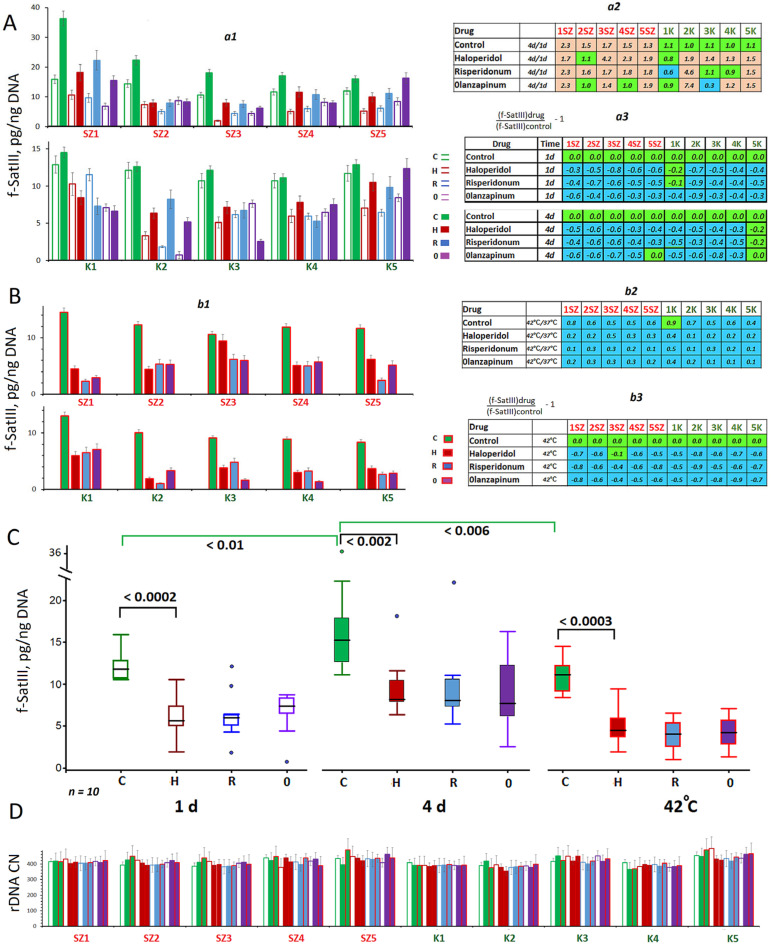
Changes in the contents of the two tandem repeats under the action of antipsychotic drugs. (**A**) (**a1**)—The f-SatIII content in DNA isolated from HSFs cultured for 24 and 96 h in the presence of antipsychotics. The average value for four measurements and the standard error are given. (**a2**)—Comparison of the f-SatIII content in one-day and four-day cultures. (**a3**)—Effect of the drugs on the f-SatIII content in DNA. (**B**) Effect of drugs on f-SatIII repeat count in cellular DNA under thermal stress (heat shock). (**b1**)—The f-SatIII content in DNA isolated from HSFs exposed to 42 °C (1 h, then 3 h at 37 °C); (**b2**)—comparison of f-SatIII content in cellular DNA before and after heating; (**b3**)—the effect of drugs on the f-SatIII content in the DNA of stressed cells. The colors in tables: green color—no significant differences from the controls (*p* > 0.01), brown—an increased f-SatIII content in the presence of the drug (*p* < 0.01) and blue—a decreased f-SatIII content (*p* < 0.01). (**C**) Comparison of f-SatIII levels in the control HSFs and exposed HSFs (drug) groups (*n* = 10). (**D**) Influence of drugs and heat shock on the rDNA content in DNA.

**Figure 2 ijms-24-11283-f002:**
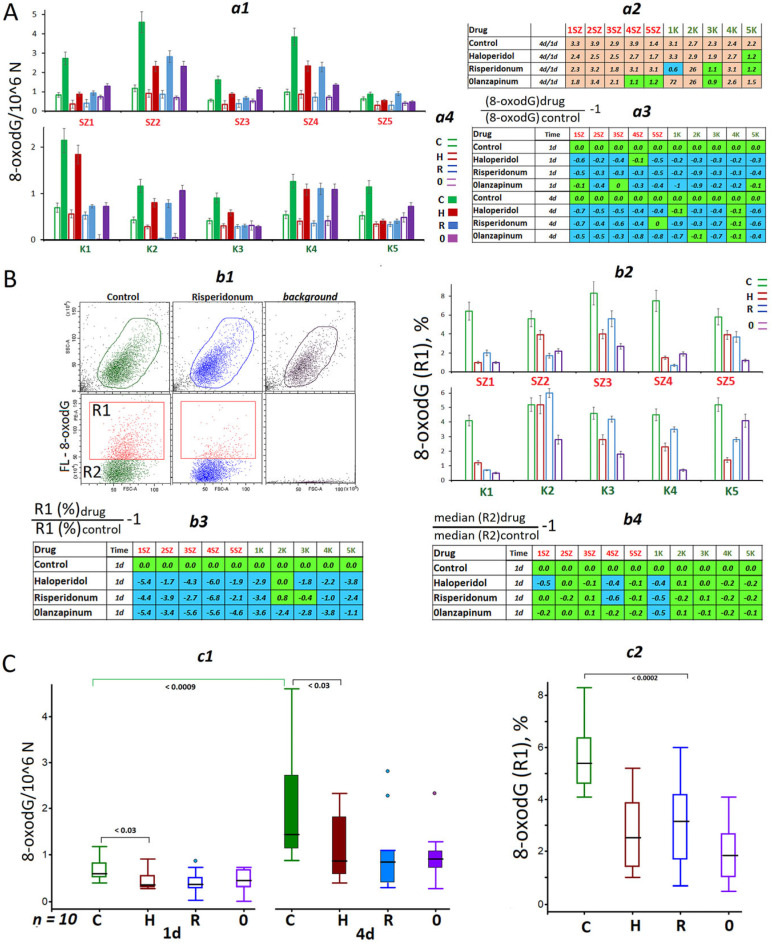
Measuring the DNA oxidation degree in the HSFs. (**A**) (**a1**)—8-oxodG content in DNA isolated from the HSFs (24 h and 96 h). (**a2**)—Comparison of 8-oxodG content for DNA of the HSFs (96 h) and HSFs (24 h) groups. (**a3**)—Effect of the drugs on the content of 8-oxodG in the DNA from HSFs. (**B**) (**b1**)—FCA example for 8-oxodG detection in HSFs. (**b2**)—R1 subpopulation (8-oxodG+) fraction for HSFs (24 h). (**b3**)—Effect of the drugs on R1(8-oxodG) fraction in the HSFs. (**b4**)—Effect of the drugs on the content of 8-oxodG in R2 subpopulation. (**C**) (**c1**)—Comparison of 8-oxodG levels in the control HSFs and exposed HSFs (drug) groups (*n* = 10). (**c2**)—Effect of the drugs on the R1(8-oxodG+) fraction in the control HSFs and exposed HSFs (drug) groups (*n* = 10). The colors in tables: green—no significant differences from the controls (*p* > 0.01), brown—an increase of the index in the presence of the drug (*p* < 0.01) and blue—a decreased of the index (*p* < 0.01).

**Figure 3 ijms-24-11283-f003:**
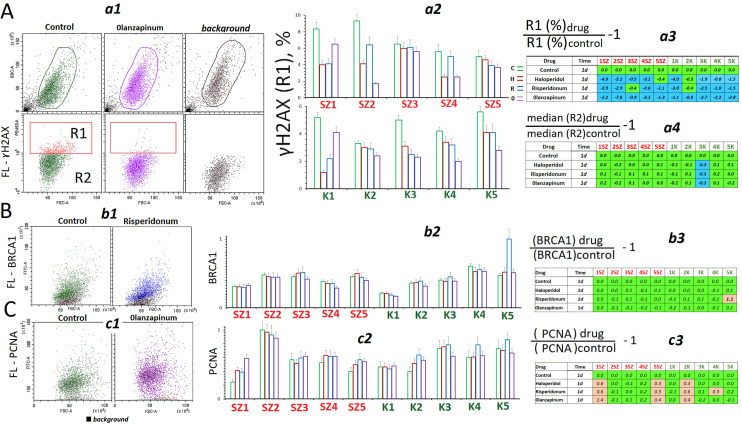
Assay of the proteins involved in DNA repair. (**A**) (**a1**)—FCA example for γH2AX detection in HSFs (24 h). (**a2**)—R1 subpopulation (γH2AX+) fraction in HSFs (24 h). (**a3**)—Effect of the drugs on the content of R1(γH2AX) in the HSFs. (**a4**)—Effect of the drugs on the content of γH2AX in R2 subpopulation. (**B**) (**b1**)—FCA example for BRCA1 detection in HSFs. (**b2**)—BRCA1 index for all HSFs. (**b3**)—Effect of the drugs on the BRCA1 levels in HSFs. (**C**) (**c1**)—FCA example for PCNA detection in HSFs. (**c2**)—PCNA index for all HSFs. (**c3**)—Effect of the drugs on the PCNA levels in HSFs. The colors in tables: green—no significant differences from the controls (*p* > 0.01), brown—an increase of the index in the presence of the drug (*p* < 0.01) and blue—a decreased of the index (*p* < 0.01).

**Figure 4 ijms-24-11283-f004:**
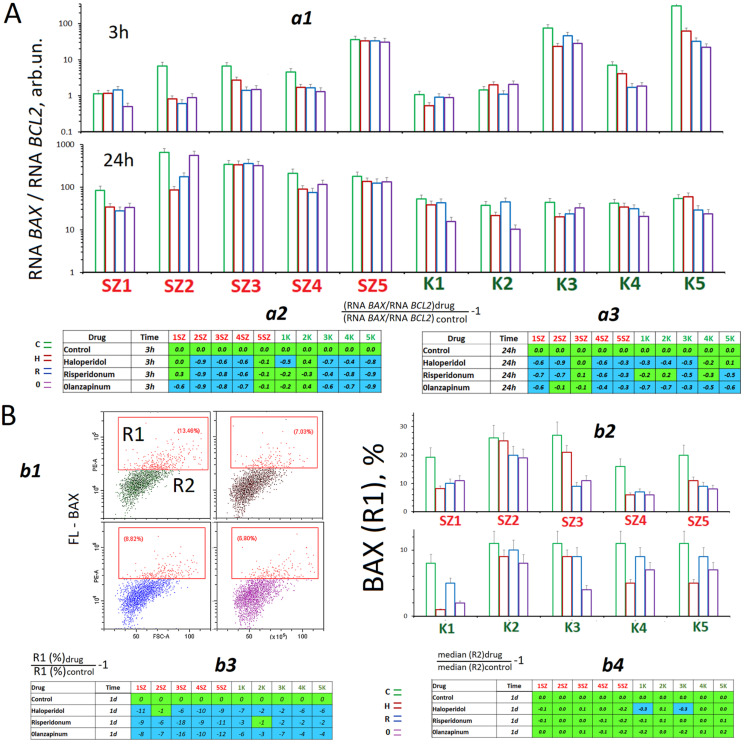
Indicators of apoptosis in HSFs. (**A**) (**a1**)—The RNA*BAX1*/RNA*BCL2* ratio in the HSFs (3 h and 24 h). (**a2**,**a3**)—Effect of the drugs on the RNA*BAX1*/RNA*BCL2* ratio in HSFs (3 h and 24 h). (**B**) (**b1**)—FCA example for BAX detection in HSFs (24 h). (**b2**)—R1 subpopulation (BAX+) fraction in HSFs (24 h). (**b3**)—Effect of the drugs on the content of R1 (BAX) in the HSFs. (**b4**)—Effect of the drugs on the content of BAX in R2 subpopulation. The colors in tables: green—no significant differences from the controls (*p* > 0.01), blue—a decreased of the index (*p* < 0.01).

**Figure 5 ijms-24-11283-f005:**
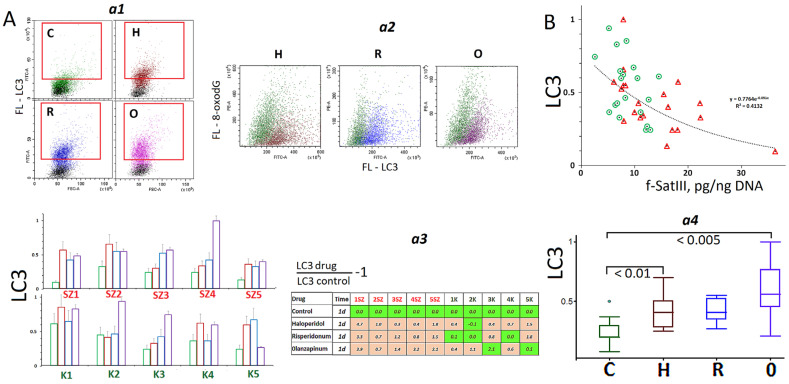
Assay of protein LC3 involved in autophagy. (**A**) (**a1**)—FCA example for detection in HSFs (24 h). (**a2**)—Determination of the marker of oxidation 8-oxodG and the level of LC3 in the same cells. (**a3**)—Effect of the drugs on the LC3 levels in HSFs. (**a4**)—Comparison of LC3 levels in the control HSFs and in exposed HSFs (drug) groups (*n* = 10). (**B**) Dependence of the LC3 level in the HSFs (24 h) on the f-SatIII content in the HSF DNA (96 h). The colors in tables: green—no significant differences from the controls (*p* > 0.01), brown—an increase of the index in the presence of the drug (*p* < 0.01).

**Figure 6 ijms-24-11283-f006:**
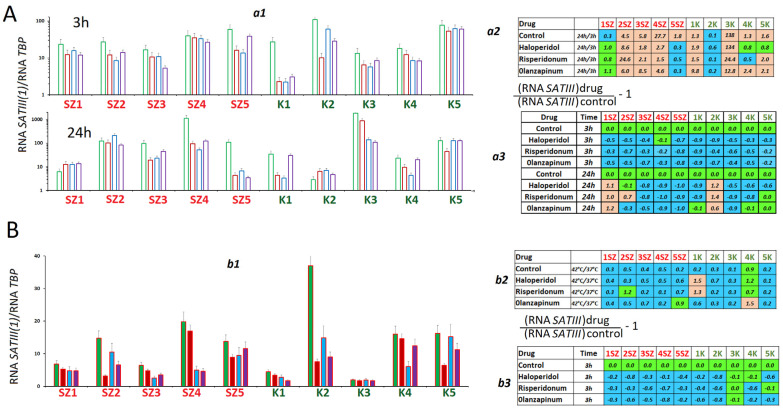
Changes in the content of RNA *SATIII* under the action of antipsychotics. (**A**) (**a1**)—The content of RNA *SATIII* in RNA isolated from HSFs cultured for 3 and 24 h in the presence of antipsychotics. The average value for three measurements and the standard error are given. (**a2**)—Comparison of the RNA *SATIII* content in RNA of one-day and four-day cultures. (**a3**)—Effect of the drugs on the RNA *SATIII* content in RNA. (**B**) Effect of drugs on RNA *SATIII* content in RNA under heat shock. (**b1**)—The content of RNA *SATIII* in RNA isolated from HSFs exposed to 42 °C (1 h, then 3 h at 37 °C); (**b2**)—comparison of RNA SATIII contents in the cell’s RNA before and after the thermal impact; (**b3**)—the effect of drugs on the RNA SATIII content in RNA of the stressed cells. The colors in tables: green—no significant differences from the controls (*p* > 0.01), brown—an increase of the index in the presence of the drug (*p* < 0.01) and blue—a decreased of the index (*p* < 0.01).

**Figure 7 ijms-24-11283-f007:**
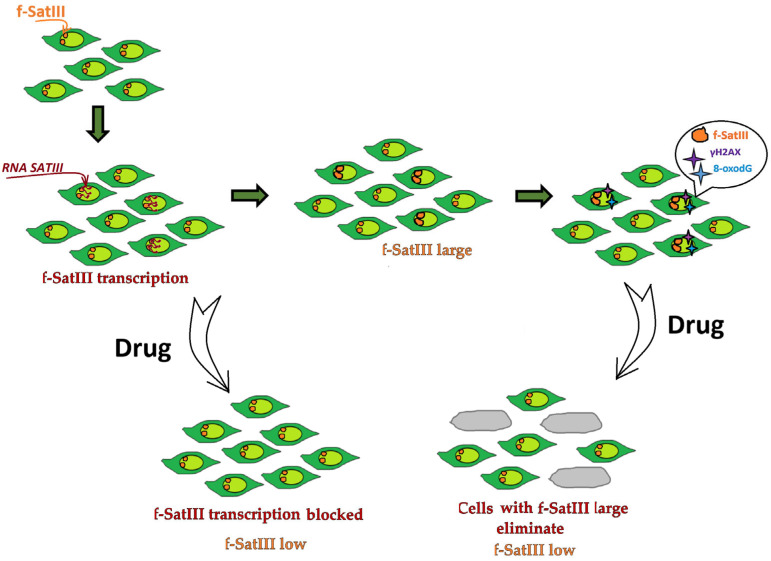
The scheme describing CNVs of f-SatIII repeats in a population of HSFs exposed to antipsychotics. An exposure to the drugs reduces the RNA *SATIII* content and activates autophagy, resulting in the elimination of the cells with high f-SatIII abundance and an elevated DNA damage degree. Both processes lead to a decrease in the fraction of cells with an enlarged block of satellite repeats.

**Table 1 ijms-24-11283-t001:** Demographic and clinical measures in the SZ patients and HC.

No.	Index	SZ1	SZ2	SZ3	SZ4	SZ5	K1	K2	K3	K4	K5
1	Age	28	39	42	50	25	35	42	49	39	27
2	Gender	m	m	m	m	m	w	w	m	m	w
3	Age of onset of SZ	14	16	17	18	12					
4	Age of manifestation of SZ	15	16	19	19	12					
5	Presence of sick relatives with SZ	+	−	+	+	+					
6	Antipsychotic treatment	+	+	+	+	+					

## Data Availability

The datasets used and/or analyzed during the current study are available from the corresponding author on reasonable request.

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
