# Peer review of "Antipsychotics Affect Satellite III (1q12) Copy Number Variations in the Cultured Human Skin Fibroblasts"

_ijms, 2023, doi:10.3390/ijms241411283_

Round 1
Reviewer 1 Report
The manuscript by Ershova et al. studies correlation between the expression / copy number variations of human f-SatIII (1q12) and antipsychotic drugs in cultured human skin fibroblast cells. This work is continuation of their long-lasting research interests, also summarized recently in the review article in Genes 2021. The main conclusion of the present work is that application of antipsychotics to skin fibroblasts reduces the copy number of SatIII, hypothesizing that the interaction via SatIII may trigger the reduction of the number of stress-damaged cells. The topic is interesting and represents a step forward in understanding putative roles and mechanisms satellites may have in pathology and in interaction with particular chemical compounds, used in the therapy. Some improvements, mostly regarding presentation, should be made for better explanation and increased readability.
In Introduction, Line 29, please explain in more details the structure of f-Sat-III. In 1q12, it is mixed with Sat-II, and the proposed mechanism of copy number increase by reverse transcriptase is based on Sat-II (ref. 19), but probably is analogous, please comment. How are they distinctive?
Line 311: The cells with enlarged block of periocentromeric heterochromatin, which harbors the f-satIII arrays… would it be possible to show the differences in size before and after the treatment with antipsychotics by FISH? If yes, I think that it would be highly beneficial and illustrative to confirm the hypotheses given.
Figures are generally overloaded, and difficult to follow without enlargements, what in turn makes difficult to follow the text. Can at least some figures be reduced to the substantial information to be shown in the main text, and the rest to present as Supplementary materials? I think that the whole Fig 1 can be in Supplement, as it shows only the best antipsychotic concentrations to be used (in Fig 1, please explain the values on x axis, number of dots on the graph, x-values are expressed as mg/ml, then as microM? Fig. 2 is the only one where the colors in tables are explained. Fig. 3., Aa2, 24 and 96 h, Aa3, 96 and 24 h; being lost. Please check if all details in all figures and figure captions are included and correctly presented. Fig. 8, could be also moved to Suppl, as the conclusion can be only approximative. The green line is mostly above the red one and increasing but it is difficult to deduce any more detailed regularity, correct?
In Fig 9, what is the meaning of red arrows, blue arrows, RNA SATIII in Italics etc? rDNA in red are not distinctive from satIII (dark yellow) in cells, please use some more distinctive colors. Is it necessary to present rDNA, as it does not change, as explained in the text? What are the brownish “bubbles” on the cells in the 2nd group? Etc. Please make distinctive and explain in detail.
Please correct / rephrase if needed: Line 353: An exposure to the drugs reduces the RNA SATIII content and >triggers autophagy off<; Line 360: We have found that antipsychotics upregulate the expression of LC3 protein, which is actively involved in the autophagy process; also in 325: the antipsychotics lower the apoptosis intensity in the cell population with (while?) inducing at the same time the process of autophagy and arrest the process of copy gain by blocking satellite repeat transcription.
There are many acronyms throughout the text, MTT test, for example, they should be explained.
Please replace thermal stress with more commonly used heat shock (stress).
Lines 120 – 125: please rewrite for better clarity. For example, I guess that this means that patients – donors of skin fibroblasts were not previously treated with antipsychotics. Acronyms as sz-HSF (and many others) should be explained upon first appearance. There are many sentences throughout the text that are difficult to follow, please revise thoroughly.
The concluding sentence at 371 is indeed a speculation, suggest to remove it.
Author Response
The manuscript by Ershova et al. studies correlation between the expression / copy number variations of human f-SatIII (1q12) and antipsychotic drugs in cultured human skin fibroblast cells. This work is continuation of their long-lasting research interests, also summarized recently in the review article in Genes 2021. The main conclusion of the present work is that application of antipsychotics to skin fibroblasts reduces the copy number of SatIII, hypothesizing that the interaction via SatIII may trigger the reduction of the number of stress-damaged cells. The topic is interesting and represents a step forward in understanding putative roles and mechanisms satellites may have in pathology and in interaction with particular chemical compounds, used in the therapy. Some improvements, mostly regarding presentation, should be made for better explanation and increased readability.
We highly appreciate the esteemed Reviewer's thorough and constructive analysis of our manuscript and positive evaluation of our modest studies. We've tried to take into account all the kind comments and suggestions.
In Introduction, Line 29, please explain in more details the structure of f-Sat-III. In 1q12, it is mixed with Sat-II, and the proposed mechanism of copy number increase by reverse transcriptase is based on Sat-II (ref. 19), but probably is analogous, please comment. How are they distinctive?
We have supplemented the manuscript as kindly suggested by the Reviewer. The Reviewer is right, the data are known for satellite II only. We assume the same or an analogous mechanism for satellite III. We are currently in the process of proving this mechanism.
Line 311: The cells with enlarged block of periocentromeric heterochromatin, which harbors the f-satIII arrays… would it be possible to show the differences in size before and after the treatment with antipsychotics by FISH? If yes, I think that it would be highly beneficial and illustrative to confirm the hypotheses given.
We previously obtained such data for lymphocytes of patients who had been analyzed before and after an antipsychotic therapy. For fibroblasts, at present, we had no time to set up an experiment and calculate it before July, 2 (the journal's deadline for replying the Reviewers' notes).
Figures are generally overloaded, and difficult to follow without enlargements, what in turn makes difficult to follow the text. Can at least some figures be reduced to the substantial information to be shown in the main text, and the rest to present as Supplementary materials?
I think that the whole Fig 1 can be in Supplement, as it shows only the best antipsychotic concentrations to be used (in Fig 1, please explain the values on x axis, number of dots on the graph, x-values are expressed as mg/ml, then as microM?
We have taken into account the suggestion. Figure 1 has been moved to Appendix 1S.
Fig. 2 is the only one where the colors in tables are explained. Fig. 3., Aa2, 24 and 96 h, Aa3, 96 and 24 h; being lost. Please check if all details in all figures and figure captions are included and correctly presented.
Figure captions have been checked and the explanations added.
Fig. 8, could be also moved to Suppl, as the conclusion can be only approximative. The green line is mostly above the red one and increasing but it is difficult to deduce any more detailed regularity, correct?
We agree with the esteemed Reviewer. Figure 8 has been moved to Appendix 4S.
In Fig 9, what is the meaning of red arrows, blue arrows, RNA SATIII in Italics etc? rDNA in red are not distinctive from satIII (dark yellow) in cells, please use some more distinctive colors. Is it necessary to present rDNA, as it does not change, as explained in the text? What are the brownish “bubbles” on the cells in the 2nd group? Etc. Please make distinctive and explain in detail.
The Fig.9 has been changed correspondingly to address each kind suggestion.
Please correct / rephrase if needed: Line 353: An exposure to the drugs reduces the RNA SATIII content and >triggers autophagy off<;
We have rephrased this line. Thank you!
Line 360: We have found that antipsychotics upregulate the expression of LC3 protein, which is actively involved in the autophagy process; also in 325: the antipsychotics lower the apoptosis intensity in the cell population with (while?) inducing at the same time the process of autophagy and arrest the process of copy gain by blocking satellite repeat transcription.
We have corrected these sentences. Many thanks!
There are many acronyms throughout the text, MTT test, for example, they should be explained.
Please replace thermal stress with more commonly used heat shock (stress).
We have corrected the wording and explained the acronym in paragraph 4.9.
Lines 120 – 125: please rewrite for better clarity. For example, I guess that this means that patients – donors of skin fibroblasts were not previously treated with antipsychotics. Acronyms as sz-HSF (and many others) should be explained upon first appearance. There are many sentences throughout the text that are difficult to follow, please revise thoroughly.
We've fixed everything. This material was in section 4.
The concluding sentence at 371 is indeed a speculation, suggest to remove it.
We agree with the respected Reviewer and we've removed the overbold sentence and replaced with a more realistic and cautious phrase.

Reviewer 2 Report
The manuscript submitted by Ershova et al. presented an interesting phenomenon that several antipsychotics, namely haloperidol, risperidone, and olanzapine, were able to cause variation in copy numbers of f-SatIII in cultured human skin fibroblasts when added in vitro. The result is both unexpected and surprising, as few investigations have reported the effect of antipsychotics in non-neuronal cells.
However, for the same reason that the experiments were conducted in non-neuronal cells, it became difficult to see the significance of the investigation and, to some degree, the relevance of the overall research. The results got more complicated as the three antipsychotics used in the study shared little pharmacological or chemical similarity. Indeed, one unavoidable question would be how the three relatively unrelated chemicals could have very similar results against the same pericentromeric heterochromatin region. Nevertheless, the results are still informative to the general audience as well as researchers specialised in heterochromatin repeat studies, and I recommend the manuscript be published as a resource for the community.
Author Response
The manuscript submitted by Ershova et al. presented an interesting phenomenon that several antipsychotics, namely haloperidol, risperidone, and olanzapine, were able to cause variation in copy numbers of f-SatIII in cultured human skin fibroblasts when added in vitro. The result is both unexpected and surprising, as few investigations have reported the effect of antipsychotics in non-neuronal cells.
However, for the same reason that the experiments were conducted in non-neuronal cells, it became difficult to see the significance of the investigation and, to some degree, the relevance of the overall research. The results got more complicated as the three antipsychotics used in the study shared little pharmacological or chemical similarity. Indeed, one unavoidable question would be how the three relatively unrelated chemicals could have very similar results against the same pericentromeric heterochromatin region. Nevertheless, the results are still informative to the general audience as well as researchers specialised in heterochromatin repeat studies, and I recommend the manuscript be published as a resource for the community.
We are grateful to the esteemed Reviewer for interesting and valuable remarks to the manuscript and especially for highly evaluating the importance of our studies.
The aim of this study was to analyze the variation of the satellite as an interesting phenomenon in itself, but not in connection with any pathology. Cells from patients were taken because the increased level of oxidative stress in this pathology had been known.
The fact is that satellite 3 increases significantly during replicative aging of normal cells and under low-level stress. Cells with a high satellite content do not respond to stimuli and are not resistant to stress. This is an unhealthy balance in the tissue. Quite by accident, earlier we found the effect of antipsychotics on the satellite content in blood cells. We thought that these compounds could reduce the number of these ballast cells in the population of any cell type. We'd like to note that all the stress reactions are very conservative and shared by a wide variety of cell types and organisms. In particular, heat shock factors are found from bacteria to human beings. The stress-derived satellite trancsription is part of the universal stress reaction, hence we have deemed it reasonable to expect that it exists in different cell types. We have tested and proved this assumption in this study on another model system.
The antipsychotic compounds have indeed a different structure (though they all contain an aliphatic cycle with a nitrogen atom), but a similar mechanism of action - block dopamine and other receptors, in addition, restore autophagy, which is impaired in schizophrenia.
What mechanisms underlies the decrease in the level of satellite transcription under the action of these compounds remains to be elucidated. However, this is a very valuable property. This is especially important for the nervious tissue where signal transmission along cell chains occurs. Blocking the functional activity of one cell due to an increase in the content of the satellite can lead to disruption of signal transmission in the chain. Perhaps a decrease in non-functional cell fraction in the brain is one of the components of the antipsychotic action.
